

# Prior experience of captivity affects behavioural responses to 'novel' environments

Edward Kluen, Katja Rönkä and Rose Thorogood

HiLIFE Helsinki Institute of Life Science, University of Helsinki, Helsinki, Finland
Research Programme in Organismal and Evolutionary Biology, Faculty of Biological and Environmental Sciences, University of Helsinki, Helsinki, Finland

## ABSTRACT

Information ecology theory predicts that prior experience influences current behaviour, even if the information is acquired under a different context. However, when individuals are tested to quantify personality, cognition, or stress, we usually assume that the novelty of the test is consistent among individuals. Surprisingly, this 'gambit of prior experience' has rarely been explored. Therefore, here we make use of a wild population of great tits (*Parus major*) to test if prior experience of handling and captivity influences common measures of exploration (open field tests in two novel contexts: room and cage arenas), social response (simulated using a mirror), and behavioural stress (breathing rate). We found that birds with prior experience of captivity (caught previously for unrelated learning and foraging experiments) were more exploratory, but this depended on age: exploration and captivity experience (in terms of both absolute binary experience and the length of time spent in captivity) were associated more strongly in young (first-winter) birds than in adults. However, there was no association of prior experience of captivity with social response and breathing rate, and nor did the measures of exploration correlate. Together our results suggest that re-testing of individuals requires careful consideration, particularly for younger birds, and previous experiences can carry over and affect behaviours differently.

## INTRODUCTION

Consistent individual variation in behavioural expression is recognised to be common across the animal kingdom with far-reaching ecological and evolutionary consequences for populations, species, and communities (reviewed in *Wolf & Weissing, 2012*). Intrinsic differences among individuals give rise to individual-level variation in traits such as exploration behaviour, aggressiveness, or risk-taking (*Réale et al., 2007*). These differences can then influence individual fitness components like survival and longevity (*Smith & Blumstein, 2008*), population dynamics and dispersal (*Dall et al., 2005*; *Cote & Clobert, 2007*; *Duckworth, 2008*), parent–offspring interactions (*McGhee & Bell, 2014*), as well as biological invasions and range expansions (*Carvalho et al., 2013*). However, the validity of measurements of animal personality depend on repeatedly testing individuals, to

Corresponding author
Edward Kluen,
edward.kluen@gmail.com,
edward.kluen@helsinki.fi

determine whether the differences in focal behavioural responses between individuals are consistent over time and context. Repeatability of so-called personality traits is often low (37% on average, *Bell, Hankison & Laskowski, 2009*), suggesting that much of the variation in individual behaviour depends on context or other environmental conditions. What contributes to this 'environmental noise' remains an open question.

Although there are calls to partition variation from a wide range of behaviours to investigate context-independent personalities and correlations among traits (*Dingemanse & Dochtermann, 2013*; *Sánchez-Tójar, Moiron & Niemelä, 2022*), many studies use a two-step approach (reviewed in *Niemelä & Dingemanse, 2014*). Here, single behavioural responses are measured under a single context before variation is related to ecological or fitness components of interest (*Niemelä & Dingemanse, 2014*), and many of the assays derived to measure putative personality traits involve capture of free-living individuals. This introduces important potential artefacts: firstly, the types of individuals captured may not necessarily be a random sample of a study population (unintended biased sampling; *e.g.*, *Biro & Dingemanse, 2008*; *Carter et al., 2012a*; *Stuber et al., 2013*), which becomes especially problematic when tests require re-capture of individuals (*Wilson et al., 1993*; *Garamszegi, Eens & Török, 2009*). Secondly, tests in captivity may generate variation in behavioural responses that might not manifest in the wild (*Niemelä & Dingemanse, 2014*) or thirdly, tests in captivity may generate variation in behavioural responses due to differences in how individuals respond to captivity itself (*Thorpe, 1956*; *Butler et al., 2006*). For instance, in a foraging behaviour experiment on wild caught chaffinches (*Fringilla coelebs*), *Butler et al. (2006)* found that birds were less successful at completing behavioural trials when the time spent in captivity prior to the tests was longer than for individuals given a shorter acclimation time. Furthermore, assays of animal personality often assume that the novelty of the test is equal for all individuals (*e.g.*, novel environment, novel object; *Carter, 2013*). This may be because many of the tests were adapted from animal psychology studies (*Réale et al., 2007*) where subjects were captive throughout their lives and thus histories were known, but this is not the case when working with wild animals.

Each of these artefacts and assumptions invokes an implicit 'gambit of prior experience' where we assume that our tests on wild individuals will be sufficiently robust to detect individual differences, despite the likelihood that individuals will not all share a similar past. Information ecology theory predicts that information gathered from a range of sources prior to an experience will influence current behaviour (*Dall et al., 2005*), and this may not necessarily vary consistently among individuals according to the personality trait of interest (*Rodriguez-Prieto, Martin & Fernandez-Juricic, 2011*; *Dingemanse et al., 2012*; *Niemelä & Dingemanse, 2014*; *Smit & van Oers, 2019*). For instance, bold trout that lose experimental fights become more shy (increased latency to approach a novel object) and shy trout that win become more bold (*Frost et al., 2007*); therefore, estimates of boldness for wild-caught trout blind to prior experience would include variation that could not be accounted for and affect conclusions. However, quantifying the prior experience of wild animals, and how it may influence their response to behavioural tests, is usually impossible. Nevertheless, as prior experience is likely to affect quantification of behavioural traits and estimates of their repeatability, it could be an issue across behavioural ecology.

Prior experience may manifest itself in the state of an individual, or be related to differences in age. However, it is well known that individuals also vary in how they respond to captivity or habituate to stimuli (*e.g.*, *Zimmermann et al., 2001*; *Carere & van Oers, 2004*; *Ellenberg, Mattern & Seddon, 2009*; *Biro, 2012*), and this may or may not reflect intrinsic variation in tolerance (*Martin & Réale, 2008*; *Allan, Bailey & Hill, 2020*). It is therefore conceivable that experiences during previous periods of captivity might affect the outcome of subsequent behavioural tests. For instance, when exploring a novel environment an individual collects information about its environment and properties (*Renner, 1990*; *Greenberg & Mettke-Hofmann, 2001*). However, the perception of the testing environment may be different for animals that have experienced captivity before. Studies on the two most commonly used inbred strains of lab-mice found that prior experience and training affected behavioural aspects differently (*McIlwain et al., 2001*; *Võikar, Vasar & Rauvala, 2004*). More specifically, mice that underwent previous handling and testing showed reduced exploration activity, emotionality and contextual fear compared to naïve mice. A previous experience of captivity may also have a carry-over effect by influencing the time taken to acclimatize to the novel environment (*e.g.*, *Butler et al., 2006*; *Martin & Réale, 2008*) and shape responses to novel stimuli. Although some attention has been given to the effects of acclimation time on a focal behaviour (*Butler et al., 2006*) and reduced response (sensitization: *Groves & Thompson, 1970*) due to habituation to behavioural tests (*Martin & Réale, 2008*; *Dingemanse et al., 2012*), these artefacts of behavioural testing methods are only beginning to be explored (*Forss et al., 2021*).

Here, we test how prior captivity influences three commonly measured behavioural traits: exploration, social response and breathing rate. We took advantage of a population of great tits, a model species for studies on animal personality, where many individuals were used previously in unrelated foraging experiments (see Table S1). Exploration behaviour is one of the most commonly tested traits in animal personality studies (*Carter et al., 2012a*) and in great tits it correlates with multiple important ecological variables (*e.g.*, dominance: *Dingemanse & de Goede, 2004*; space use: *van Overveld & Matthysen, 2010*; *van Overveld, Adriaensen & Matthysen, 2011*). Exploration behaviour is assayed most often using a novel arena in a room (*e.g.*, *Dingemanse et al., 2002*) or cage (*e.g.*, *Stuber et al., 2013*), and so here we used both types of novel environments to estimate repeatability (*Carter et al., 2012a*) while reducing the potential for habituation and learning effects (*e.g.*, *Réale et al., 2007*; *Uher, 2011*). To assay social response behaviour we used a mirror test that provides a perfectly matched competitor and responses can vary from avoidance to attraction. Mirror tests originate from research on primates but is increasingly used to quantify behaviour in birds (*e.g.*, *Henry et al., 2008*; *Carvalho et al., 2013*). Finally, we measured breathing rate to quantify response to handling stress (*e.g.*, *Carere & van Oers, 2004*). Breathing rate has been shown to correlate with exploration behaviour and boldness in great tits (*Carere & van Oers, 2004*; *Fucikova et al., 2009*) and cage activity in blue tits (*Cyanistes caeruleus*, (*Kluen, Siitari & Brommer, 2014*)). All behavioural assays were novel, but nearly half of the birds had some prior experience of captivity and handling. We predicted that birds with prior experience of captivity would be less stressed by handling and more acclimatised to the testing environment (*i.e.*, have a slower breathing rate), and thus more exploratory

and responsive to a conspecific. If prior experience had an absolute effect on behaviour, we expected an association with a binary measure of captivity experience (*i.e.*, yes/no), or if the effect of prior experience increased linearly then we expected an association with the number of days spent in captivity.

## MATERIALS & METHODS

### Study site and catching methods

During March 2018, we caught 50 great tits around the University of Jyväskylä Konnevesi Research Station (62.616°N, 26.346°E) using permanent feeder-traps baited with peanuts and familiar to the birds as they are open throughout the winter (see *Ham et al., 2006* for details). An additional three great tits were caught over two days in the city of Jyväskylä (62.253°N, 25.750°E), using the same type of peanut trap. Birds were transported to the laboratory within 10 min after trapping, or within two hours when caught in the city of Jyväskylä, and all birds were caught between 9am (sunrise at 06:40am) and 12pm. We also used a commercially available 'tent spring trap' (TW45 Moudry-trap; http://www.moudry.cz) baited with seeds at additional locations around the research station for a different study on blue tits and caught a further three great tits as by-catch to use in this study (total sample size $N = 56$). While trapping location was recorded, the trap style used for each bird was not. However, we checked if differences in trapping location affected our results by repeating analyses with the three birds from Jyväskylä omitted (Table S2). Results were qualitatively the same so we retained them in our dataset.

### Housing

Birds were weighed (mass to the nearest 0.25 g using a 30 g Pesola (http://www.pesola.com) spring balance) upon arrival and released into individual plywood housing boxes (Fig. 1A, 65 cm length × 60 cm width × 80 cm height) where they were in auditory, but not visual, contact (as used during previous experiments, *e.g.*, *Hämäläinen et al., 2020*). The housing boxes had a perch, were illuminated by a light bulb (for no more than 12 h per day), and the front consisted of a sliding darkened Plexiglas wall that facilitated observation by observers but limited visual contact for the bird as the boxes were in a dark room. All birds had access to drinking water and food *ad libitum* (mix of peanuts, sunflower seeds and commercially available fat granules) and were left undisturbed for at least one hour before the first behavioural assays commenced. All behavioural assays for each bird were performed during one day (11am–6pm) and took place during natural daylight hours. Once the assays were completed, individuals were measured (wing length to the nearest mm and mass as above), ringed with a numbered metal ring issued by the Finnish Museum of Natural History (if un-ringed) and sex and age (young (first winter)/old) were recorded based on plumage characteristics (*Svensson, 1992*). Birds were released during daylight hours at the site of trapping before 4pm; when birds could not be released before 4pm, birds were housed overnight and released the following morning. All animal care was provided or supervised by a highly-trained animal care technician employed by the research station, and no birds were harmed or euthanised during the experiment.

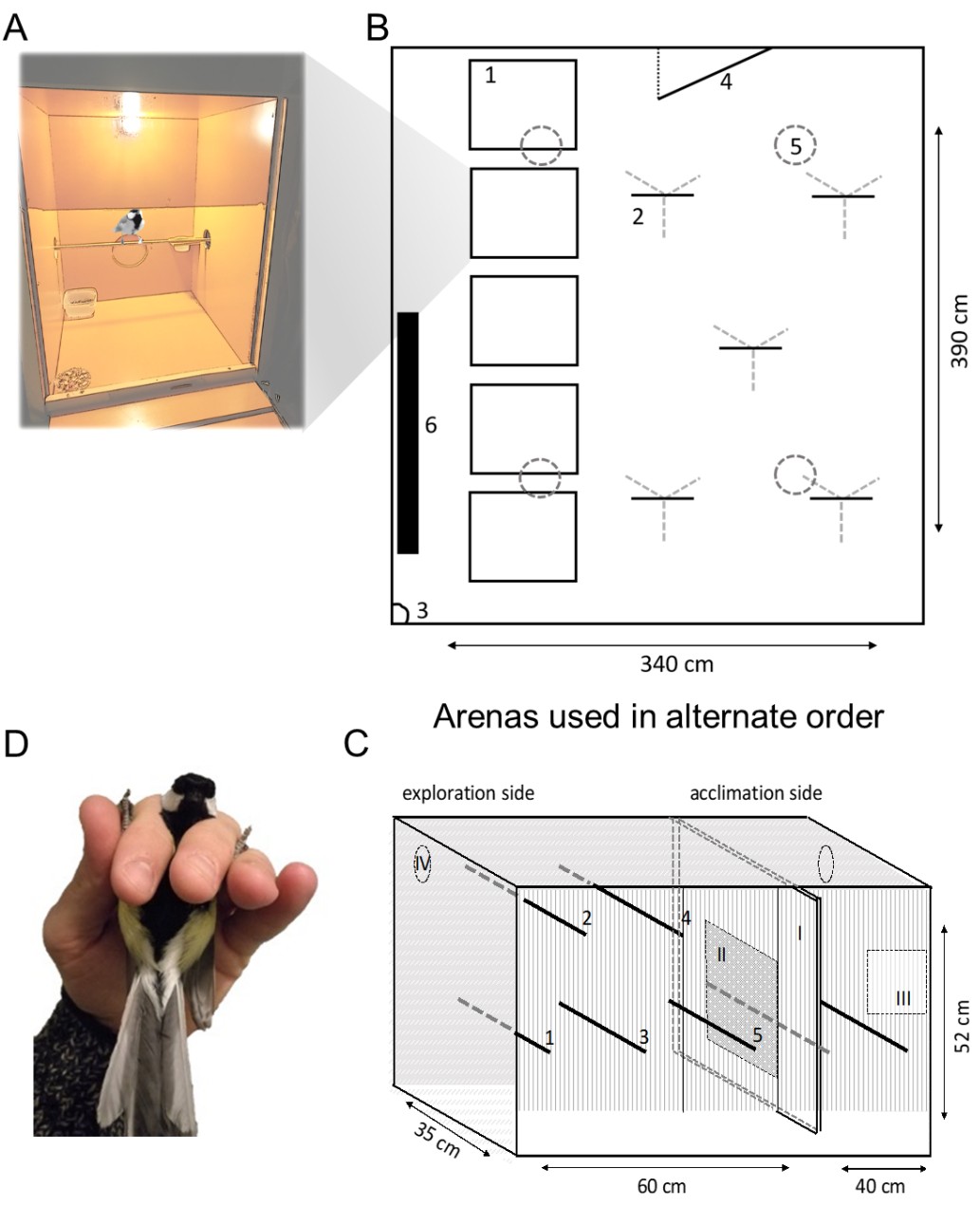

**Figure 1  Schematic of behavioural assays.** Great tits were housed in (A) boxes before and between behavioural assays. Exploration behaviour was assayed twice using: (B) a room arena (plan view) and (C) a cage (frontal view). In (B), birds were released individually from (1) housing boxes (A) to explore (2) five artificial 'trees' (metal tripod with wooden perches). Exploration was recorded by (3) camera and observation through a one-way window in the (4) door; birds could land on (5) ceiling lights but not (6) wall radiator (not in use). In (C), birds were placed in the 'acclimation side' before (I) sliding door was opened 10 cm to allow movement in the 'exploration side' (five perches, numbered 1–5) illuminated by (iv) two LED lights (not used as perches). A (II) mirror was exposed to assay social response behaviour before the bird was recaptured via a (III) door and (D) breathing rate recorded in the 'ringers grip'.

## Capture history

Almost half (26 out of 56: 46%) of the trapped birds had been held previously in the research facilities (from here on referred to as 'captivity experience'). They experienced similar housing and handling procedures as described above and participated in a range of learning experiments where they were offered food items of varying palatability in plywood boxes similar to their housing box (Table S1). Birds were caught semi-randomly with respect to captivity experience, although experienced birds tended to be caught more often earlier in the study (Fig. S1) as these were needed for another experiment. Date was therefore included as a covariate in all analyses (see data analysis methods).

## Exploration assays

The behavioural assays performed in this study were novel to all birds. Exploration behaviour was measured twice for each bird within one day, once in a room and once in a cage. The order of these two trials was alternated and the birds were left undisturbed for at least one hour in their housing box with food and water *ad lib.* between trials. The exploration trials and social response assay were recorded and videos analysed using open-source software 'BORIS' (*Friard & Gamba, 2016*).

The room was intended to match as closely as possible to the layout described in detail by *Dingemanse et al. (2002)*, given logistical constraints and existing architecture (Fig. 1B). The room measured 390 cm length × 340 cm width × 230 cm height, and contained five artificial trees that were 150 cm tall, each with four 20 cm long horizontal 'branches'. At least one hour prior to the start of the exploration assay, each bird was brought from its housing box in its original resting place using a cotton bird bag and housed individually in one of the five boxes (similar in layout and size to their original housing box) placed on the floor along one side of the room. The fronts of the five housing boxes were covered in dark plastic to prevent visual contact with the room. Each trial was filmed using a remote controlled GoPro camera (model: Hero4 Session (GoPro, San Mateo, USA)) fixed to an upper corner of the room for later analysis, see Video S1 for an example video clip of room exploration, while an observer followed the bird through a one-way window in the door. Immediately prior to the start of the trial, an observer entered the room in complete darkness, switched off the home box light and opened the sliding front wall of the focal individual's house box. The observer then left the room and switched on the room light remotely, marking the start of the trial. Each bird was individually tested, where the focal bird was given 10 min to enter and explore the room while the observer scored its movements through a one-way window in the room's door, see below for how the exploration behaviour was scored. If a bird started to explore only after eight minutes of the trial, we continued observations until two minutes of exploration had been completed. Once the trial ended, the bird was coaxed to return to its home box by the observer switching off the lights in the room and switching on the light in the box.

Following *Dingemanse et al. (2002)*, we recorded the number of flights and hops a bird made during the first two minutes of exploration in the room (*i.e.,* when a bird emerged from its housing box), through the one-way window. All flights and hops within a 'tree' (but not within a 'branch'), and between 'trees', floor, ceiling or walls were counted. One

bird hopped mainly on the floor of the room, this individual was scored according to approximate 'between tree distance' covered, and 10 individuals did not explore during the assay and thus were assigned a zero score for room exploration in the analysis.

Our second assay of exploration behaviour used a cage arena (Fig. 1C) adapted from previous studies with great tits (*Stuber et al., 2013*) and other passerines (*Herborn et al., 2010*; *Kluen et al., 2012*; *Zhao & Sun, 2016*). The cage (100 cm length × 35 cm width × 52 cm height) was placed on a table in a dark room (a battery-powered led light at both ends of the cage provided illumination), and all sides were opaque except for the front which consisted of metal mesh to allow filming. A sliding wall (double layer, see mirror assay) divided the cage into two compartments: an acclimation side (40 cm length × 35 cm width × 52 cm height) containing two perches and an exploration side (60 cm length × 35 cm width × 52 cm height) containing five perches. Each layer of the dividing wall could be moved using a plastic rod attached to the wall, meaning the observer remained out of view of the bird. Each bird was tested individually. The assay was filmed using a Canon PowerShot A2400 (Canon, Tokyo, Japan) camera mounted on a tripod 150 cm from the cage, see Video S2 for an example video clip of cage exploration. Immediately prior to the trial, the bird was caught from its housing box (within one minute of switching off the home box light) and released into the acclimation side of the cage for five minutes. Filming of the trial started immediately after the bird was released. After five minutes, both layers of the sliding wall were opened simultaneously by 10 cm, giving the bird 10 min the opportunity to move into the exploration side of the cage.

From the video we recorded the number of hops and flights for two minutes once the bird emerged from its acclimation side into the exploration side (either by choice or when coaxed through after 10 min, see above). All flights and hops between (but not within) the perches, floor or mesh were counted. Ten birds (32%) returned to the acclimation side during the two minutes. For these birds, we decided to include all movements made, regardless of which side of the cage the bird was in, rather than negatively bias their measures of exploration. This was because the majority (nine out of ten) spent more time on the exploration side (percentage of time spent in the acclimation side; one bird: 2%, seven birds: 21–30%, one bird: 41%) and visits to the acclimation side were in short bouts of a few seconds (except for one bird who returned to the acclimation side for 74% of the two minutes in one 88s bout). All birds showed continuous movement on the acclimation side, exhibiting similar behaviour to that on the exploration side. Three birds entered the exploration side of the cage just before the end of the 10 min assay and were erroneously assumed (during the assay) to have been present for 2 min before the social response assay began. Therefore, the exploration time for these birds was very short (22 s, 28 s and 46 s) and they had a shorter time than other birds to familiarize with the cage before exposure of the mirror. We omitted these three birds from all analyses. Twenty-two birds did not enter the exploration side of the cage within 10 min. These birds were assigned a zero score for cage exploration. The birds that did not explore the cage were coaxed into the exploration area and were given five minutes to explore this area, with the sliding doors closed to prevent return to the acclimation side, before the social interaction assay was

performed, to allow familiarization with the exploration area of the cage. In general, these coaxed birds were all very inactive, where movements in 2 min ranged from 0–3 moves.

## Social response behaviour assay

Immediately after the cage exploration trial, we conducted a mirror test to assay social response behaviour following the methods of *Carvalho et al. (2013)*. The exploration-side layer of the sliding wall was removed to expose a rectangular mirror (17.5 cm × 25.0 cm) attached to the (closed) acclimation side wall layer and positioned at the same height as three of the five perches (Fig. 1B). Birds could not see their reflection from the other two perches as they were above the mirror (*i.e.*, due to the 'law of reflection'). Behaviour towards the mirror was filmed for five minutes before the lights were switched off and the bird was caught from the cage by hand. See Video S3 for an example video clip of response towards the mirror.

From the videos we recorded: the number of pecks against the mirror and the time a bird spent looking at the mirror (time spent looking at the mirror was counted when the bird's beak was at maximum of 90 degrees angle towards the mirror, following *Carvalho et al. (2013)*, the amount of time (in seconds) birds spent on each perch and the number of movements (in the same manner as for the exploration assay). We then calculated the proportion of time spent on each perch as a percentage of the total time spent with the mirror. These values were used to calculate the difference in time spent on each of the five perches relative to the time spent on these perches before the social challenge (based on *Kluen et al., 2012*) using the following formula:

$$\log[(X + C)/(Y + C)]$$

where $X = $ proportion of time spent on a perch during mirror exposure trial, $Y = $ proportion of time spent on the same perch during exploration trial, $C = 1$ was included to account for log-transformation of zero values. The proportion of time spent on each perch during the exploration trial was relative to the total amount of time a bird spent on the exploration side of the cage (*i.e.*, not including any time spent back on the acclimation side). For birds that were coaxed to the exploration arena of the cage, we used the five minutes after coaxing to calculate this measure. The same formula above was used to calculate difference in number of movements but instead of proportion of time on each perch, we used movements per second. A score of zero for both relative time spent on a perch and movements indicated no change in behaviour and positive values indicated that the bird spent more time on the perch, or was more active, during the mirror exposure than during exploration.

## Breathing rate

Immediately after removing the bird from the cage, we recorded its breathing rate in the hand following *Kluen, Siitari & Brommer (2014)*. The bird was held with its back on the palm of the observer's hand and its neck between the observer's index and middle fingers (*i.e.*, ringers grip, Fig. 1D), with one leg held between thumb and index fingers and the other by the middle and ring fingers. We then measured the time it took a bird to take 30

breaths (fall and rise of the chest) twice, using a stopwatch. Breathing rate was calculated as the average of these two measurements and expressed as the number of breaths per second.

## Data and statistics

### Principal Component Analysis (PCA) on social response assay

We used a Principal Component Analysis (PCA) to reduce the number of variables extracted from the social response behaviour videos. To improve the principal component interpretation we used data on two of the five perches in the PCA using the following reasoning; The birds' own mirror reflection could not be seen from two out of the five perches, due to the angle (perches high in the cage). Time spent on these two perches was negatively correlated ($r = -0.59$) and we opted to choose only one of these perches in the PCA *i.e.*, the perch furthest away from the mirror. Furthermore, from the three perches at mirror level, we opted to leave out the two perches furthest from the mirror as these contained many zero's (*i.e.*, they were not often used during the assay). The other variables in the PCA were: 'proportion of time spent looking at the mirror', 'change in number of movements (compared to exploration)' and 'number of pecks at the mirror'. The first Principal Component (PC1; 41%) represented interaction with the mirror: number of pecks on the mirror, relative time spent on the perch closest to the mirror, and increased movement were positively loaded while relative time spent on the perch furthest from the mirror was negatively loaded (see for PC1 loadings). Subsequent PC's (PC2: 24% & PC3: 23%, Table S3) were dominated by time spent looking at the mirror (PC2) and increased movement (PC3), however these were not used in further analyses as PC1 already included these behaviours (time spent looking at the mirror was loaded positively but weakly; Table S3). We used the scores of each bird on PC1 as the behavioural variable for the social response assay in further analyses.

### Phenotypic correlations among behaviours and associations with prior experience of captivity

We first explored phenotypic correlations between the different behaviours using a Pearson's correlation tests. The two assays of exploration behaviour were included separately (*i.e.*, exploration in the room and exploration in the cage). Next, we tested for associations of prior experience of captivity and behaviours using multivariate mixed models (MMMs). MMMs are generalised linear models where multiple response variables are modelled using both fixed and random effect terms. We did this in a Bayesian framework, using R-package MCMCglmm (*Hadfield, 2010*). We ran 3,050,000 iterations, used a burn-in of 50,000 and sampled the chain every 3,000 iterations, ending up with 1,000 samples of the chain. We used an inverse gamma uninformative prior and, as we only had repeated measures for exploration, we set within individual variance and within individual correlations for all other behaviours to zero following *Houslay & Wilson (2017)*. Convergence of the chains was checked using plots of the MCMC samples. Exploration behaviour consisted of count data (number of movements in 2 min); therefore, we used a Poisson error distribution in its model and a Gaussian distribution for all other behavioural variables (*i.e.*, PC1 and breathing rate). We ran our analyses twice to investigate whether the length of captivity (*i.e.*, continuous predictor, days in captivity, scaled to the mean and unit

standard deviation) or experience of captivity (*i.e.*, binary predictor, yes/no) influenced each behaviour. To assess whether any relationships were significant, we used the estimated coefficients with their 95% Bayesian Credible Intervals (BCI). We considered a significant effect for those covariates which credible intervals did not overlap zero. As a random effect in both models, we used 'bird identity'. The fixed effects structure for both models was the same, with the exception of the captivity variables, and included the following: 'body mass' (scaled to their mean and unit standard deviation), 'sex' (male/female), 'age' (first winter/older) and 'catching date' (mean centered and scaled by the standard deviation). We also included an interaction between age and each variable describing prior experience of captivity to account for variation in life experience and the possible impacts of developmental processes underlying the expressed behaviour (*e.g.*, *Stamps & Groothuis, 2010*). When estimating effects on exploration behaviour we included two additional variables: 'test-type' (cage or room, to account for differences in the results due to the arena) and 'test order' (whether conducted first or second). In addition, we checked whether 'test-type' influenced the effects of variables describing prior experience of captivity (assessed using an interaction term in a univariate model; Table S4) but as it was not significant, we did not include it in the MMMs. This indicates that any association of captivity experience with exploration behaviour is not dependent on the arena in which exploration was tested.

Finally we estimated the repeatability of our tests for exploration behaviour, conditional on the fixed effects (adjusted repeatability), using the random effects (co)variances from the MMM (following *Houslay & Wilson, 2017*) where prior experience of captivity was included as a continuous measure (*i.e.*, days in captivity). We estimated repeatability by dividing the between-individual variance estimate by the sum of the between-individual and residual variance and used the credible interval to assess significance. All statistical analyses were performed in R version 3.4.3 (*R Core Team, 2017*).

## Ethics

All experiments complied with Finnish law on animal experimentations. Permits for the capture and use of great tits in experiments were granted by the Central Finland Centre for Economic Development Transport and Environment (ELY; VARELY/294/2015) and licensed from the National Animal Experiment Board (ESAVI/9114/04.10.07/2014). Bird trapping was done only under dry (no precipitation) weather conditions. Bird handling was done by skilled persons and with highest possible care and all birds were trapped and released (at the trapping location) during natural daylight hours.

## RESULTS

After excluding 3 birds with erroneous exploration measurements (see 'Methods'), our dataset included assays conducted with 53 great tits (Table S5), of which 26 had experienced captivity before, ranging from 2–29 days spent in captivity spread out over 1–4 capture events (see Fig. S2). Our dataset included 24 females and 29 males; for the age of the birds 32 birds were in their first winter and 21 were adults.

**Table 1  Phenotypic correlations between the three behavioural variables.** Phenotypic Pearsons correlations and their 95% confidence intervals between each of the behavioural variables measured on 53 great tits ($N = 53$). Where high PC1 values represented interaction with the mirror: number of pecks on the mirror, relative time spent on the perch closest to the mirror, and increased movement (compared to movements during exploration) were positively loaded while relative time spent on the perch furthest from the mirror was negatively loaded.

|  | Exploration - cage | Exploration - room | Breathing rate |
|---|---|---|---|
| Exploration –cage | – |  |  |
| Exploration –room | 0.04 (−0.23–0.31) | – |  |
| Breathing rate | −0.17 (−0.42–0.10) | 0.00 (−0.27–0.27) | – |
| PC1 (interaction with mirror) | 0.19 (−0.08–0.44) | 0.24 (−0.03–0.48 ) | −0.14 (−0.40–0.13) |

**Phenotypic correlations between the behavioural variables**

Surprisingly, the two measures of exploration were not correlated, nor were any of the behaviours significantly correlated with one another (Table 1). However, both exploration measures tended to correlate positively with PC1 (*i.e.*, more exploratory birds spent more time close to the mirror and pecked against it) although these correlations were not significant (Table 1). We acknowledge that, to detect low significant correlations we should have a large sample size and that ours may not be sufficient in doing so.

**Does prior experience of captivity influence exploration behaviour?**

We found that both the number of days spent in captivity and binary experience of captivity were positively associated with exploration behaviour (Table 2, Figs. 2A & 2B). This relationship was independent of the type of test used following from the interaction of days in captivity and test-type and experience of captivity and test-type (Table S4). However, the effect of prior experience of captivity on exploration behaviour did depend on age where older birds with experience of captivity were less exploratory than younger birds that also had previous experience of captivity following the interaction of age and the binary variable of captivity (Table 2 and Fig. 2B). Furthermore, for birds without previous experience of captivity, older individuals were more exploratory than younger ones (Fig. 2B). This interaction with age was also significant for the continuous variable of captivity (Table 2 and Fig. 2A), and was not driven by the 10 older birds with a longer captivity experience than the younger birds (results not shown). Test-type was significantly associated with exploratory behaviour, where birds explored less during the first two minutes in the cage than in the room (Table 2). Days in captivity and experience with captivity were not associated with social response behaviour (Table 2, Figs. 2C & 2D) or breathing rate (Table 2, Figs. 2E & 2F). However, sex was significantly associated with breathing rate, where males breathed slower than females (Table 2). In addition, there was a tendency that heavier birds breathed faster following from the nearly significant association of body mass and breathing rate (Table 2).

**Is exploration behaviour repeatable?**

The estimated adjusted repeatability of exploration behaviour was 0.12 (BCI 95%: [<0.01–0.34], Table 2) and the lower credible interval value was very close to zero, indicating that the repeatability value was not significant.
**Table 2  Multivariate generalized linear mixed effects models (MMMs) of prior experience of captivity on behaviour.** The fixed effect coefficients ($\beta$) and random effect parameters ($\sigma^2$) with their 95% credible intervals (CI) based on a multivariate mixed effect model on three behaviours in 53 great tits and the adjusted repeatability of exploratory behaviour based on model 1. (i) Model 1 included prior experience of captivity as a continuous variable (Days in captivity); Coefficients from a similar multivariate mixed effect model(ii) Model 2 where experience of captivity was included as a binary variable (0/1). Significant effects (credible intervals not crossing 0) are given in bold. The fixed and random effect coefficients of model 2 are presented in Table S6.

| | Exploratory behaviour moves in 2 min $N = 53$ | Breathing rate breaths $s^{-1}$ $N = 53$ | Social interaction –PC1 (Close to and pecking at mirror, movements (+)) $N = 53$ |
|---|---|---|---|
| **Fixed effects** | $\beta$ **(95% CI)** | $\beta$ **(95% CI)** | $\beta$ **(95% CI)** |
| Intercept | **1.75 (0.87–2.79)** | **2.37 (2.22–2.52)** | **−0.52 (−1.34–0.21)** |
| Bodymass | 0.05 (−0.46–0.47) | 0.09 (0.00–0.18) | −0.30 (−0.68–0.12) |
| Sex (male) | 0.45 (−0.36–1.35) | **−0.28 (−0.45−−0.12)** | 0.50 (−0.28–1.35) |
| Age (old) | 0.09 (−0.97–1.04) | −0.05 (−0.23–0.13) | 0.72 (−0.26–1.56) |
| Catching date | 0.41 (−0.14–0.99) | −0.05 (−0.15–0.05) | −0.21 (−0.69–0.32) |
| Test type (cage) | **−0.91 (−1.69−−0.14)** | – | – |
| Test order | −0.43 (−1.14–0.30) | – | – |
| **(i) Model 1: continuous measure of captivity** | | | |
| Days in captivity | **1.50 (0.54–2.60)** | −0.08 (−0.27–0.09) | 0.27 (−0.71–1.06) |
| Days in captivity * Age (old) | **−1.46 (−2.60−−0.35)** | 0.03 (−0.15–0.22) | −0.24 (−1.12–0.77) |
| **(ii) Model 2: binary measure of captivity** | | | |
| Experience of captivity | **2.15 (0.78–3.79)** | −0.04 (−0.31–0.24) | 0.44 (−0.81–1.95) |
| Experience of captivity * Age (old) | **−2.07 (−3.88−−0.15)** | −0.13 (−0.48–0.21) | −0.30 (−2.07–1.38) |
| **Random effects** | $\sigma^2$ **(95% CI)** | | |
| Bird_ID | 0.50 (<0.01–1.41) | | |
| Residual | 3.30 (1.94–5.03) | | |
| | **Link-scale approximation R (95% CI)** | | |
| **Repeatability (adjusted)** | 0.12 (<0.01–0. 34)[a] | | |

**Notes.**

[a]Because the variance components are constrained to be positive in MCMCglmm models, a lower bound of the credible interval close to zero indicates low confidence in a non-zero proportion of the phenotypic variance in exploratory behaviour being explained by differences between individuals.

## DISCUSSION

Here we took advantage of knowledge regarding wild individual great tits' capture histories to examine whether the 'gambit of prior experience' may affect point measures of behaviour. We predicted that birds with prior experience of captivity would be more likely to explore and interact with a competitor (mirror image) because they were more habituated to the general context of our testing environment. Our results supported these predictions to an extent: prior experience of captivity influenced exploration behaviour, but this effect varied according to age. Younger birds exhibited a stronger response to previous experience of captivity, exploring more when they had spent more time in captivity before or had ever been caught. Adult birds, on the other hand, showed limited variation in their exploration behaviour in response to prior captivity. We found no difference in social response behaviour or breathing rate, suggesting either that captivity has little effect on social responses or stress, or that intrinsic responses remain consistent across

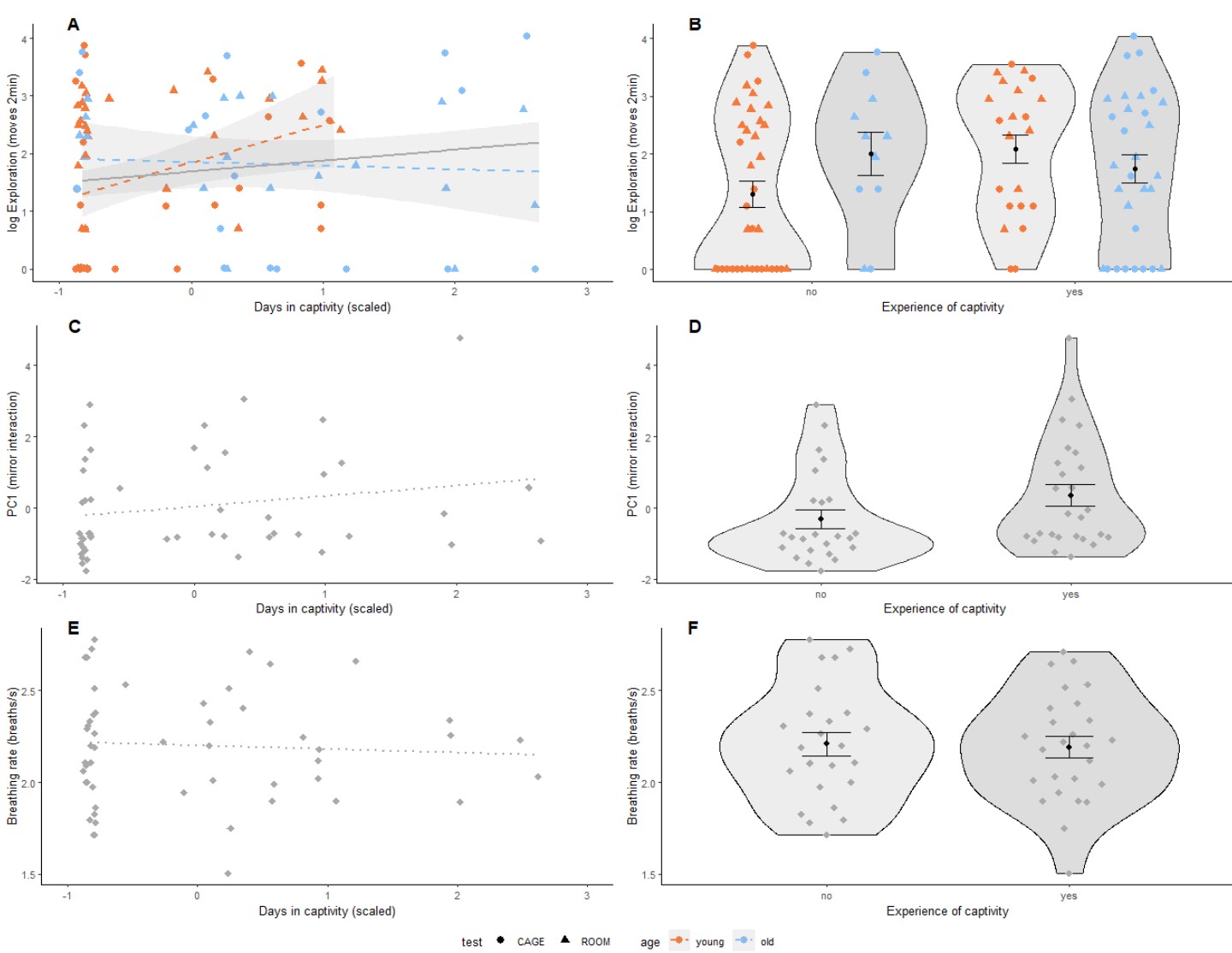

**Figure 2  Prior experience of captivity and behaviour in great tits.** (A, B) Exploration behaviour (shown on log-scale as Poisson distributed) was measured in two arenas (room: triangle, cage: circle); (C, D) Social response behaviour was measured using a mirror, where PC1 represents inter-action (number of pecks on the mirror, and time spent on the perch closest to the mirror and movements relative to behaviour during cage explo-ration assay); and (E, F) Breathing rate (breaths per second), $N = 53$ birds for all panels. Prior experience of captivity is shown as a continuous vari-able (mean centered and scaled by standard deviation) in (A, C & E), and as a binary variable in (B, D & F). The association of prior experience with captivity and exploration behaviour differed by age (*i.e.*, significant interaction, Table 2) so data points and regression lines in (A & B) are indicated accordingly (first-winter: orange, adult: blue). Non-significant regressions are shown as dotted lines. In (B), (D), & (F), black dots indicate group means with standard error bars. Note that the interaction in (A) remains if adult birds with more than 12 days in captivity (scaled days in captivity greater than 1.5) are removed.

capture events. The latter is plausible for breathing rate, as this measure is often found to be repeatable (*Fucikova et al., 2009*; *Kluen, Siitari & Brommer, 2014*). Together our results suggest that re-testing of individuals requires careful consideration, particularly for younger birds, as a single event of captivity may be sufficient to affect the outcome of a behavioural

measurement, and previous experiences might carry over and affect behaviours differently. This could be especially important when the behaviour of interest is related to novelty.

Previous work has shown that exploration of a 'novel' environment scales with repeated experience of the same test (*e.g.*, *Minderman et al., 2009*; *Dingemanse et al., 2012*), yet here we show that experience with captivity in general also has an effect on exploration. This relationship may occur through two not mutually exclusive pathways. First, repeated capture may lead to habituation, the lessening of a response after repeated exposure to the same stimulus (*Thorpe, 1956*). In our case, habituation to prior experience of captivity could have enabled these individuals to explore more actively because they were less stressed (*e.g.*, *Baugh et al., 2013*) and therefore assessed the risk associated with the captive situation differently (*Augustsson & Meyerson, 2004*). However, we did not detect any correlation between breathing rate (*i.e.*, stress, *Carere & van Oers, 2004*) and exploration, or between breathing rate and prior experience of captivity, but stress measured in the hand may capture a different aspect of stress than that experienced during the acclimatization and exploration periods of our tests. Capture bias, a widely recognised issue in animal behavioural testing (*Biro & Dingemanse, 2008*; *Carter et al., 2012a*; *Biro, 2012*), provides an alternative explanation. Here, bolder, dominant or more aggressive individuals may be more likely to be re-captured at feeding sites, whereas shyer and less aggressive individuals may avoid these after having experienced capture. In this way, naïve birds may be more random with respect to behavioural types. Hence, unintended biased sampling may have occurred and capture history may be confounded. However, capture bias alone is unlikely to explain our results as we also found an effect of length of captivity. Furthermore, multiple individuals in our dataset were caught only once but held in captivity for longer than birds caught multiple times (Fig. S2) and when we checked our key result, we found no relationship between the number of times captured and exploration behaviour (Table S7).

The effect of prior experience of captivity on exploration behaviour may also reflect variation in how birds value acquiring new information. It is notable that here we found young birds explored more actively compared to adults with similar experience of captivity, including both in terms of binary experience of captivity and the length of their experience. In some studies, young birds continue to sample information about their surroundings, whereas adults rely more on previous experience (*e.g.*, *Aplin, Sheldon & Morand-Ferron, 2013*; *Franks & Thorogood, 2018*; *Franks et al., 2020*; *Hämäläinen et al., 2021*), and young birds may therefore be more flexible and continue to explore despite prior experience of captivity (but see *Penndorf & Aplin, 2020* for a meta-analysis showing a lack of age effects in foraging behaviours). In addition, developmental processes play a large role in the expression of behaviour across context and time (*Stamps & Groothuis, 2010*). For instance, a study on ravens (*Corvus corax*) and crows (*Corvus corone, Corvus cornix*) found that exploration diminishes over ontogeny (*Miller et al., 2015*) and changes in exploration over age (juvenile/adult) in great tits were due to changes in the underlying genetic architecture of exploratory behaviour during development (*Class, Brommer & van Oers, 2019*). Similar age context-dependent effects were detected in great tits in terms of dominance and exploration

(*Dingemanse & deG oede, 2004*), providing further evidence that age in combination with prior experience may affect behavioural responses differently.

While we cannot yet explain why prior experience with captivity influenced exploration, our results do provide support that using single measures of behavioural traits may be unreliable. Like many studies, we measured exploration twice across two measurement contexts, however, like some other studies (*Burns, 2008*; *Fox et al., 2009*; *Stuber et al., 2013*; *Kluen & Brommer, 2013*; *Fisher et al., 2015*; *Arvidsson et al., 2017*), we found only mixed evidence that the measures were repeatable. Accounting for capture experience only further reduced repeatability (Fig. S3). This is because capture experience is associated with exploration behaviour and therefore any variation in individual experience with captivity will add to differences between individuals in exploration behaviour. Admittedly, our sample size was smaller than recommended to be able to detect a low repeatability (*Dingemanse & Dochtermann, 2013*), but this is a common problem (*Dingemanse & Wright, 2020*). Single behavioural observations mainly reflect residual variation (reviewed in *Niemelä & Dingemanse, 2018*; *Dingemanse & Wright, 2020*) coming from various contextual or environmental sources, and as our results of prior experience with captivity indicate, this may also be context-dependent according to age.

Alternatively, methodological differences between the two test setups may underlie the lack of repeatability; our two exploration arenas may have been testing different aspects of behaviour. This too is a common problem in studies of animal personality (*Carter et al., 2012a*). While some researchers use the same setup multiple times in the same contexts (*e.g.*, *Dingemanse et al., 2012*), others urge that measures should be done in different setups to ensure the validity of a trait (*Carter et al., 2012b*). In our experiment, we followed the latter approach using a room and a cage for exploration, however using the cage introduced differences in the way we handled the birds. Birds tested in the room were housed in their home boxes and released from these boxes into the exploration room, whereas the birds tested in the cage were moved by hand from their home boxes and given a five minute acclimation period before the exploration trial began. It is possible that a component of the exploration behaviour that we measured in the cage was actually a response to handling. Nevertheless, both of our measures of exploration tended to correlate with social response behaviour (mirror test) in the same direction and strength (albeit marginally non-significantly). This finding is comparable to a previous study (*Moiron et al., 2019*) where exploration behaviour of great tits measured in a cage was weakly correlated with territorial aggressive responses to a taxidermic mount of a male conspecific. Altogether, this stresses the importance of taking into account contextual differences between behavioural measurements, but also differences in prior experience of the tested individuals.

## CONCLUSION

In conclusion, our results highlight an issue in studies of behavioural traits that has thus far been largely overlooked, the gambit of prior experience. Using individual histories of captivity, we found that the effects of prior experience on exploration behaviour varied with both the age of individuals tested and the behaviours measured. Indeed, our results suggest

that tests commonly associated with novelty are likely to be most affected. Although we do not yet know the mechanisms underpinning the effects of prior experience on exploration behaviour, our study highlights that habituation to captivity may affect acclimatisation and information use, and furthermore, these effects may be context-dependent on both age and testing methodologies. This could be especially important when designing behavioural experiments, as from an ethical (the three R's; Replacement, Reduction and Refinement *Russell & Burch, 1959*) and economical point of view there is a need to reduce the number of animals used in experiments, and studies of animal personality often emphasise the need to recapture and retest individuals to derive appropriate measurements. Our results show that when making decisions on whether to use naïve individuals or reuse individuals, it is important to establish whether previous, and perhaps seemingly unrelated, experience may affect behavioural measurements in subsequent trials. In wild animals, it is usually impossible to ascertain prior experience of tested individuals. However, we suggest taking prior experience into account wherever possible.

## ACKNOWLEDGEMENTS

We are grateful to the staff at Konnevesi Research Station for providing facilities for the experiment and in particular Helinä Nisu for taking care of the birds. We are thankful to Johanna Mappes and members of her research group that shared their data to compile the captivity histories of the birds in our study (Table S1).

### Funding

This research was supported by a start up grant to Rose Thorogood from HiLIFE Helsinki Institute of Life Science. Open access funded by Helsinki University Library. The funders had no role in study design, data collection and analysis, decision to publish, or preparation of the manuscript.

### Grant Disclosures

The following grant information was disclosed by the authors:
Rose Thorogood from HiLIFE Helsinki Institute of Life Science.
Helsinki University Library.

### Competing Interests

The authors declare there are no competing interests.

### Author Contributions

- Edward Kluen conceived and designed the experiments, performed the experiments, analyzed the data, prepared figures and/or tables, authored or reviewed drafts of the article, and approved the final draft.
- Katja Rönkä conceived and designed the experiments, performed the experiments, authored or reviewed drafts of the article, and approved the final draft.

 

- Rose Thorogood conceived and designed the experiments, performed the experiments, analyzed the data, prepared figures and/or tables, authored or reviewed drafts of the article, and approved the final draft.

## Animal Ethics

The following information was supplied relating to ethical approvals ({i.e.}, approving body and any reference numbers):

Permits for the capture and use of great tits in experiments were granted by the Central Finland Centre for Economic Development Transport and Environment (ELY; VARELY/294/2015) and licensed from the National Animal Experiment Board (ESAVI/9114/04.10.07/2014).

## Data Availability

The data and R-code is available at Zenodo:

Edward Kluen, Katja Rönkä, & Rose Thorogood. (2022). Data and R-code belonging to Manuscript: ''Prior experience of captivity affects behavioural responses to 'novel' environments'' [Data set]. Zenodo. https://doi.org/10.5281/zenodo.6865934.

## Supplemental Information

Supplemental information for this article can be found online at http://dx.doi.org/10.7717/peerj.13905#supplemental-information.

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
