# Peer review of "Prior experience of captivity affects behavioural responses to ‘novel’ environments"

_PeerJ, doi:10.7717/peerj.13905_

## Round 0.1 · original submission · Major Revisions

Thank you for submitting this study to PeerJ. I regret that I am unable to accept the manuscript for publication, at least in its present form. However, I am prepared to consider a new version that carefully takes into account the suggested revisions. The reviewers liked many aspects of your study but also highlighted some concerns. These need to be addressed in detail in the new version. Such a revised manuscript may be reviewed again and there is no guarantee of acceptance. When you revise your paper, you should prepare a detailed explanation of how you have dealt with all of the reviewers' and Editor's comments.

·

Basic reporting

I thank the authors for their work. The manuscript is clearly written and enjoyable to read. However, PeerJ publishes articles in American English whereas this manuscript uses British English. I advise the authors to make the necessary edits so as to follow the journal’s guidelines.

I noticed a few errors throughout the Reference list. Three citations in your manuscript cannot be found in the references: Dingemance et al. (2013) L50; Class, Brommer & van Oers (2019) L438; Kluen & Brommer (2013) L450. Conversely, Meehan & Mench (2002) is mentioned in your references (L681) but is nowhere in the main text. Finally, I advise the authors to stick to one way of writing the cited authors’ names (e.g., Réale). Apart from that, the article provides relevant literature.

Figures and tables’ descriptions need more detail. To cite only two examples, I suggest that you make sure “PC1” is always defined and that you always give the “n”.

I thank you for providing the raw data as well as supplemental files. I noticed some minor typos.

Experimental design

I found the topic of this study very interesting. Discussing the various biases that come with the study of animals’ personality, cognition, or stress is of major importance in order to produce high-quality studies.

The studied population of great tits seems to be a suitable model for this work. The hypotheses are clear and in tune with the proposed title.

Figure 2. How did you scale the “days in captivity”? I don’t understand how there can be a [-1 ; 0] part. Moreover, you have data about “old” birds that experienced more than 1 day of captivity but not on “young” birds. I do not recall it being mentioned somewhere. Isn’t the comparison between the two categories biased because of that?

Validity of the findings

L312-314. Same idea written three consecutive times. Moreover, it is clearly stated that significance is attained when the interval does not overlap zero. You say that the effect of bodymass on breathing rate is significant (L.375 and Table 2) whereas your interval includes zero?

L358-377. The results here focus on the interaction between captivity experience and exploration behavior. You state that older birds with experience of captivity were less exploratory than younger birds with similar experience. However, in Figure 2, there is no mention of this result?

·

Basic reporting

In general, the overall structure and flow is okay. However, some of the writing is not clear and can be frustrating to find the necessary information. I am also confused and lost with some of the results and statements, some of them were contradicted with each other. The formatting can be improved as well.

Please see the PDF file for details.

Experimental design

Please see the PDF file for details.

Validity of the findings

Please see the PDF file for details.

Additional comments

I find the topic interesting and potentially important to the field. The idea that the authors trying to investigate here sounds solid in theory, and very likely to influence animal behaviour studies afterwards.
However, even though this is a preliminary work to investigate the idea, I have major concerns regarding multiple aspects of this study, from the study design, execution, analysis, to the findings and conclusions drawn.
I genuinely hope the comments could help the authors to improve this work, and get this idea publish. Definitely looking forward to it.
A reject is recommended for this manuscript.

Reviewer 3 ·

Basic reporting

The paper "Prior experience of captivity affects behavioural responses to 'novel' environments" examines whether prior experience with captivity influences commonly used measures of exploration (open field test in two different settings), social response (mirror test) and behavioural stress (breathing rate). Overall, I found it to be an interesting addition to our understanding of how such often omitted factors such as prior captivity experience, might influence behavioural measures. The manuscript was well structured, and arguments are well formulated and easy to follow, making it an enjoyable read.

Experimental design

Overall, the design is well explained and decisions concerning the analysis well explained. I do however have some comments/ questions.

L270-271: I am not certain if I understood the stated formula. As it is written, I understand it as follows:
log (x +C/y+C) where x is the proportion of time spend on a perch during the mirror exposure trial, and y the proportion of time spend on the same perch during the exploration trial. Since you state C=1 (L273), could you please explain why you take the inverse of the proportion of time spend on the same perch during the exploration, but the proportion of time in the mirror exposure trial ?


Concerning the models, I enjoyed seeing a Bayesian approach. While not being an expert in the use of Bayesian analyses, I do however have some concerns about the way the analysis was carried out. First, the use of an uninformative prior (inverse gamma uninformative prior) is, to the best of my knowledge, to be avoided. Rather, priors should be chosen in regard to the distribution of the data. Second, I would have liked to see the posterior distributions of the models in the supplementary material as these would allow us to see the fit of the models, and thus the validity of the findings.

Validity of the findings

Given the concerns I emitted in 2. concerning the analysis, I am at current not able to judge the validity of the findings.

Additional comments

I stumbled across a few typos, I list them here in the hope that it will be of use.
L131: missing ponctuation after "behaviour"
L312-314: repetition of the same sentence

Reviewer 4 ·

Basic reporting

No Comment

Experimental design

The methods require some re-structuring to improve clarity and ease of reading. No other comments.

Validity of the findings

No comment.

Additional comments

Dear Editor
Thankyou for inviting me to review the MS “Prior experience of captivity affects behavioural responses to 'novel' environments”. The authors look at how prior experience of captivity (in terms of a binary yes/no and the number of days in captivity), is associated with exploration behaviour, breathing rate (as a measure of stress) and response to the social mirror test. They find that prior experience in captivity affects exploration behaviour but not the other two behavioural measures. This affect was influenced by age of the birds.

The study is very well written and put together and it was a pleasure to read. The introduction flows well and the arguments are nicely structured. I don’t believe there is any major literature or discussion points that are missing. I believe the study is a useful contribution to the literature as it touches on a very important point in behavioural ecology when studying wild animals, how their prior experience, that we can never know everything about, may influence their performance on behavioural tasks in captivity, especially when repeated measures are required.

I do not have very many comments to make, only a few restructuring notes and reconfiguring of figures, please see below.

Raw data check: The raw data and code can be opened and is well described with notes for the variables and detail notes in the code throughout.

Image check: Figures and images have not been inappropriately manipulated.

English Language check: The level of English is excellent. No change in language is required.

Vertebrate animal usage check: There is an ethics statement and it seems wholly appropriate. Experiments were necessary and birds were ethically kept.

Major points:
- Restructure of the methods. The video analysis section can be incorporated into the other sections where each behaviour is talked about separately. This will be clearer than having a separate section.
- Figure 1 should be changed and separated. There are panels included that are most likely better in the supplementary. And the supplementary illustrative figures are better in the main text. See my minor points.

Minor points:
Introduction:
Line 121: It is not clear why you have looked at social response, until you mention further down your predictions. I think just a sentence of explanation on the reasoning behind this should be included in the beginning part of this paragraph.

Methods:
Figure S1: The abbreviation for ‘number’ on the y axis is not standard. Either write the whole word or put ‘No.’.

Figure 1: It is not clear what panel A of Figure 1 is showing. There are three colours on the graph but only two colours in the legend. Is the X axis the exploration score in two minutes? It seems to me that you want to show the distribution of birds that participated, but you are also trying to show their score…? This is not clear and the score has not been mentioned in the text yet when we first come to look at Figure 1 so should be explained in the legend. Is the Y axis showing the number of birds that participated? The axis labels should be made clearer and the legend is too brief. More detail is needed to explain exactly what we are looking at. The photos in B are not the best quality so it is better to have the illustration? The photo of the cage is particularly difficult to see what the bird is doing. Your Figure S2 is a good representation of the layout of the test and is more informative than the photos so perhaps is better as the main figure rather than the supplementary.
Photos D and F are nice and illustrate the point well.

Line 183: I have a few questions about logistics of the boxes: Were the housing boxes physically moved from their original resting place into the exploration room? And in the room were they placed on the floor? Why were five boxes moved into the room at once?

Line 188: When I first read this, I wasn’t sure if you recorded the exploration score for the whole ten minutes or just the final two minutes at the end of the ten minutes. I think mentioning the ten minutes here is a little confusing. Make it clear that you recorded exploration behaviour for two minutes only. Why did you give the birds ten minutes in total?
You could add here about the scoring as well: either “see below for how the exploration behaviour was scored.” Or “The number of hops and flights in two minutes was recorded, see below section on video analysis”.

Line 189: I assume birds were tested individually but you should clarify this. This also relates to my above question about why there were five housing boxes in the exploration room at a time.

Figure S3: This plan is very useful and would be better in a main figure rather than a supplementary. The photo of the cage would probably be better in the supplementary if you want to include it.

Line 210: Again, is the exploration score calculated within the first two minutes of the bird entering the testing side of the cage? And why are they given ten minutes total? You should also mention here briefly how you scored the exploration score (and if it was the same as the larger test in the room) or refer the reader to the video analysis section below. It is nice when reading the set-up of the room and the cage to be able to know how the bird was scored, rather than having to wait until reading about the video analysis further down.

Figure 1: In panel C, the PC1 is mentioned but when 1C is first mentioned in the text we have not yet come across PC1, so this does not follow very well. Either rename the x axis to something else or make very clear in the legend what PC1 is. Again, expand the legend and make it clear what the graph is showing. I’m not clear really why you have these graphs in your main Figure 1 because they do seem to be referring to results when you cite them first in the methods. If you mean for them to be results (even results of distributions) then I think they would work better as a separate figure (in the supplementary would be best) and referred to in the results section of the text. Figure 1 I think should be the illustrated plans of the room and cage tests.

Line 224: I think here we need a bit more information about how the response to the mirror was scored. I know you refer to a paper with the methods but the reader should be able to understand how the test was conducted and how it was scored without referring to that other paper.

Line 235: I see why you have video analysis as a separate section but I had a number of questions about how the exploration scores and the mirror test were measured, that you then explain in this section. So I would restructure the methods slightly and integrate the video analysis part into the above sections. Have one section for each behavioural trait: exploration behaviour (room and cage), social response and breathing rate, and you can state the full methods for each section. The Data and Statistics section is fine to be separate.

Line 285: Instead of ‘multiple’, ‘number’ is better: “To reduce the number of social response variables…”

Line 290: ‘table’ should be capitalised and Table S4 should be in italics for consistency with the rest of your ms.

Line 308: Firstly, I think here you mean to refer to Figure 1C and 1E. Secondly, are you using panel C and E to justify why you used a gaussian distribution? E does look normally distributed but I would not say that the data in C is normal. It has a heavy skew. This is the first specific reference to these graphs and as I mentioned above, I think they should be in a separate figure and referred to first here in the statistics part of the methods and put in the supplementary information.
You say you ran your analysis twice with length of captivity or act of captivity, but why could you not put these two explanatory variables in the same model? I may have misunderstood about the MMMs.

Line 312: You have a repeated sentence here. And the following sentence on line 313 seems to be saying the same thing again…?

Results:

Table S3: Perhaps saying ‘captivity history’ would be better than saying ‘experience’. It took me a little while to comprehend the fact that although you say you included the test type * captivity interaction in both models, it didn’t seem like it was included for Model 2. When in fact, you’ve given it a different name. I would always use the word captivity. Saying ‘experience’ is slightly confusing.
Table 2: Same applies here with ‘experience’.

Figure S4: there is a typo in ‘relative’ in the second line.
There seems to be two outliers here (one caught once and one caught four times). The one caught four times may be driving the effect?

Line 349: the numbers given here for the correlation between the two exploration measures are not quite the same as those given in Table 1. Please check both. Same for PC1 and cage on line 353.

Table 1: Please explain PC1 in the legend.

Figure 2: Figure legends should not have a title (same applies to Figure 1). Integrate the title into the body of the legend and describe as clearly as possible, and with detail, what the figure shows.
A suggestion is to make the different exploration tests (cage and room) in different shaped symbols in this figure. The different colours work well for the age and instead of having different symbols also for age, you could make the cage trials circles and the room trials triangles, for example.
Again as in Figure 1, describe more what the figure shows. i.e. on the first line you say ‘exploration behaviour measured in two contexts’ but this doesn’t describe what A and B show. You also don’t say what the contexts are. Try something like: “Exploration score of the great tits in relation to (A) the number of days they have been in captivity, and (B) the binary option of whether they have experienced captivity before”. Try the same for the other behaviours. Explain more of what PC1 shows.

Line 365: Instead of saying ‘similar’, change to: “…experience of captivity were less exploratory than younger birds who also had previous experience of captivity”. Also, from Figure 2B it seems that there is a big difference between young and old birds that do not have experience of captivity. So if this is the case then you could add in this aspect too and say something like “In birds with no previous experience of captivity, older ones were more exploratory than younger ones.”

Line 393: instead of saying “…showed little response…” maybe change this to “…showed limited variation in their exploration behaviour, in response to captivity”. I think this sentence would benefit from mentioning the exploration behaviour specifically.

Line 418: “… after having experienced capture”.

Figure S5: There is a typo in ‘relationship’ on the first line. You could say “The relationship between the number of times a bird was caught and exploration behaviour”.

Line 411: Just a thought: Instead of being mediated through stress, could it not just be that they are more used to the captive situation and so are less stressed....? And are more curious about moving around their environment? I suppose this very much depends on how stress manifests in the individual: does it cause hyper activity (flight) or lower activity (freeze).

Line 441: what about the idea that perhaps juveniles are more likely to be able to adapt to their habitat because they are younger. So the previous experience with captivity influences their current state more than adults that are already fairly fixed in their behaviour. Juveniles are more flexible and adults are more fixed in their behaviour.

Line 444: This doesn't seem quite right to me. You were not measuring whether replications of the same trait were correlated or not. You show that previous experience in captivity influences exploration behaviour which is slightly different. I think this sentence needs revising. At the very least explain here how your results provide support for this idea.
You go on to say that because you have found that captivity experience influences exploration, single measures may be context dependent... but I do not think your results provide support for this idea... simply that this is an idea that might be inferred from your results.

Line 449: Change to: "...exploration across two contexts". Don't say measured twice because it sounds like you measured twice in each context.

Line 459: Alternatively to what?

Line 478: You don’t need to say ‘additional’ in this sentence.

Line 480: “… we found that the effects of prior experience on exploration behaviour vary with…”

R code: Check your text carefully. I found a small error in line 50 where you say that Table S4 shows the PC1 loadings when I think you mean Table S2.

---

## Round 0.2 · Major Revisions

Thanks for the careful revisions to the previous version of this manuscript. As you can see from the new reviewer comments, considerable editing is still required to get the manuscript to the required standard for the journal. Please be extra careful with the new revisions. Apologies for the delay in getting this decision back - it resulted from the extra time needed to get the new version reviewed.

·

Basic reporting

The English language is still really good. I only noticed some back and forth between American and British English.
I also have some other minor comments:

I advise the authors to make sure they spell out numbers when they are between zero and nine.

I think it would be more enjoyable if the authors stick to one way of writing units, either min/h or minutes/hours for example. Similarly, the authors should stick to one way of mentioning long x wide x high (both in the manuscript and figures).

L124-126. I believe this sentence should either be "To assay social response we utilised a mirror test. This test provides a perfectly matched competitor and responses can vary from avoidance to attraction" or "To assay social response we utilised a mirror test that provides a perfectly matched competitor and responses can vary from avoidance to attraction".

L196-200. I believe it should be “At least one hour prior to the start of the exploration essay, each bird was brought from its housing box in its original resting place using a cotton bird bag and individually housed in one of the five housing boxes […]”.

I apologize for not reporting it in my first review, but I believe it is common to mention the manufacturer of the different devices used, e.g., L202 GoPro Hero 4 Session (GoPro, San Mateo, USA) and L233 Canon Power Shot A2400 (Canon, Tokyo, Japan).

Although it is better than the previous Figure 1, I personally find the legend of the (new) Figure 2 hard to read.

I thank the authors for providing the raw data as well as supplemental files. I could open everything. However, in supplementary material, careful your first URL is incorrect. Make sure : is before //. It should be https://doi.org/10.5281/zenodo.6483295

Experimental design

L150-151 you say N=56 birds. L377 N becomes 53. In the M&M I understood that you were using data from all the caught birds? Did I miss something?

Validity of the findings

The other reviewers pointed out some concerns about the original manuscript. The authors seem to have taken them into account and changed their work accordingly.

Additional comments

I thank the authors for their work. I believe they answered the reviewers’ comments and that the manuscript needs some minor edits before Acceptance.

·

Basic reporting

I urge the authors to check and proofread the manuscript carefully for cross-referencing and inconsistencies in reporting style. The mislabeled Figures and mismatched info are causing this manuscript unreadable at some point. I do not have enough time to screen and pick them all out in here but do hope the authors could double check the manuscript.

Please go through the number one to nine in text, as the PeerJ guideline indicates they need to be spelt out unless go with units. Although I am not sure “1 minute” counts as with unit or not. Authors use both “1 minute” and “one minute” in text, along with other inconsistencies in reporting.

L117: “Table S1” is okay, no need to add “in the supplement”.

L142: Konnevesi “R”esearch “S”tation, use capital letters for places.

L142-154: The catching date/period info is missing. In previous version, this info was available: March 2018.

L147: “2h” should be written in full as two hours or 2 hours. And Jyväskylä was referring the city or the research station? It is not clear here.

L147-148: “sunrise at 06:40 am and 12 pm” the time reporting format is not unified.

L162: “observation by us…” the word “us” seems a bit weird. Authors could consider using “observers” or so, just like in L206.

L167: “wing to nearest mm and mass as above” - I cannot be sure the word “wing to” is a typo or not, I am raising this just in case. It is weird if only measure the wings.

L196: “long*wide*high” here should be written as “length*width/depth (depends on the context)*height”.

L197: Here the authors mentioned each artificial tree in the room arena was “with four 20 cm long horizontal branches”. However, in Figure 2, L6-7 indicated the trees were “metal tripod with wooden perch, 150 cm h x 40 cm perch”. Please align this piece of info.

L267: “acclimation-side” should be without the hyphen.

L281-282: For easier reading, the authors could consider the one mentioned by Reviewer 3, which is:
Log[(x+C)/(y+C)]
Then, could list the follow explaining the equation:
x = proportion of time spent on a perch during mirror exposure trial
y = proportion of time spent on the same perch during exploration trial
C = 1 (A constant for log transformation of zero value)

L304: Principal “C”omponent “A”nalysis, go for capital letter as it is a specific term.

L379: The use of brackets here seems inappropriate, the info inside the brackets is at the same level of importance. One or two simple sentences should do here. For example, “Among the 53 Great Tit studied, 29 were males and 24 were females; for their age, 32 were first winter and 21 were adults.

L381: An extra full stop for title. Please delete it.

L393-394: The result figures listed here cannot be found in Table 2 or any other tables, both main tables and supplementary tables. Please crosscheck the results. L401-402, 406-407, and 415 are not matching with any of the Tables as well.

L395-397: The results here are from the univariate model only and not the MMMs. It is confusing to refer to both Table 2 and S4 here.

L427: The word “either…or…” seems confusing. Both binary and continuous measure were significant based on the results, so should not be only one factor stands?

Figure 1 & 2: Please check the figures’ number and in text references. All of them are currently mismatch or mislabeled. This makes the manuscript hard to follow or even unreadable.

Figure 1 (the A-F result graphs):
Figure legend L5: “breaths second-1“ should be written as breaths per second.
Figure legend L6-7: I noticed the authors’ reply to Reviewer 1 (R1.5) saying added the details of scaling the days of captivity in both main text and Figure 2. However, I cannot find it in the main text after a few quick scouts and using search function.
Figure legend L11-13: I cannot locate this info anywhere in the main text. And why choosing 12 days / scaled days greater than 1.5 as the threshold?

Figure 2 (the schematic one):
1) Too many details for the figure legend, readers can find the procedure in the Methods section. Simply explain or list out the symbols and/or numbers used in the figures should be fine. For example, in B(i): 1 is black boxes; 2 is artificial trees; 3 is the camera…etc. The authors could look at the example given in the PeerJ guideline.
2) Figure legend L8-10: This info cannot be found in the main text.
3) Subfigure A: The edited photo is not clear. The black box layout should be in the same style as B(i) and B(ii).
4) Subfigure C: I do not think this is needed. And the picture is not clear with the effect.

All Tables (both in main text and supplementary): “N: 53” should be written as “N = 53”, please go through all the Tables again.

All in-text references and reference list are all green, no issues found.

Supplementary: The authors could consider rearranging and grouping the supplementary info by their nature: Table, Figure or Video. The current order based on the appearance order is hard to follow.

Experimental design

No specific comment.

Validity of the findings

L152-154: By addressing only 3 individuals were from a different location should be okay and cleared the questions regarding the possibility of population level differences influence the results. Excluding the analysis (Table S2) in the manuscript is okay to me. An article with 7 supplementary tables is too bulky in general.

I am not familiar enough with the methods performed here, even though I have used GLM before, I am not capable to provide any further comments. However, I have two questions regarding the analysis still:
1) Giving the resemblance of the two captivity factors (binary and continuous), is it possible for the authors to consider using only one for the MMMs for better communication, and using correlation to show the continuous form?
2) Another area is regarding the test type, the MMMs indicated the test type had main effect on its own (not talking about the interaction terms here). The correlation also suggested the exploration behaviour from the 2 types were not in the same direction, which birds were not just consistently explored less in the cage and more in the room. I acknowledged the authors’ reply in the rebuttal letter, and I agree with the authors only if the two explorations were correlated.

Additional comments

I would like to thank you the authors addressed the reviewers’ comments in detail and all their hard work, especially the additional analyses. The work and details are now clearer and less confusing than the pervious version in certain aspects.
And again, the idea investigated by the authors here is potentially important to the field. However, mainly due to the comments I mentioned in the Basic Reporting above, it is difficult to recommend accepting this manuscript for publication at this stage.

Reviewer 4 ·

Basic reporting

No comment

Experimental design

No comment

Validity of the findings

Please make sure that all the results in the text match the tables.

Additional comments

The authors have done a good job of responding to my initial comments. There are however some things that still need to be changed to make the paper publishable.
Main points:
- Results – the numbers in the text do not match the tables. Make sure everything is accurate. Check the journal guidelines but I think you don’t have to put the numbers in the text. You can just put the variable or the interaction and then the table to refer to for the reader to find the numbers.
- I am confused by the distinctions between table 2 and s4. Table 2 is the multivariate model results and you state that table s6 is the results for model 2 (still multivariate). So where do the fixed and random effect estimates in table 2 come from? Also, table s4 is the univariate effects and you say are testing the effect of captivity*test type, and yet these two interactions are also present in table 2. Clarification is needed on why you have done the univariate analysis.
- Really check the whole MS for inconsistencies in type (e.g. italics) and how tables and figures are written, as well as units.

Line 26: Reads better to say “in terms of both…” rather than “both in terms of…”. Then it’s clear that the ‘both’ refers to the binary experience and the length.
Line 27: Should read ‘of’ rather than ‘to’? “…and the length of time spent…”
Line 60: instead of the semi-colon here, you could just put a comma and then say ‘which becomes especially problematic…”
Line 63: ‘artefacts’ is not really clear. Rephrase perhaps to say ‘tests in captivity may generate variation in behavioural responses due to differences in how individuals respond to captivity itself’.
Line 81: This sentence is fairly long and you say ‘differences between individuals’ twice within it. Try and be more concise.
Line 84: may work in a similar way to what?
You don’t really need to say ‘often’. ‘Repeated’ will do.
Line 86: This sentence and the previous one both say ‘this’ and I think by this point we have lost what ‘this’ refers to. Make the subject of the sentence clear. I think here you mean to say ’prior experience generates behavioural differences...’ You don’t even need to say ‘may’ if there are studies to show that it does.
Line 93: ‘grossly’ doesn’t add anything here.
Line 104: after ‘differently’ you need the references cited.
Line 117: Table S1 is fine. Don’t need to say ‘in the supplement’.
Line 153: Table S2 here in italics. Double check the whole ms for these inconsistencies.
Line 158: Figure 1 here is the images of the cage, the illustration of the room and showing the bird held with the breast stripe. But this does not correspond with what you’ve put as figure 1 on the pdf version. Please be very careful and double check that everything matches correctly.
I assume here you mean: 65cm x 60cm x 80cm; L x W x H. Make sure you put the LWH in some form. You’ve written ‘high’ here but you should include length and width too. I would just use L x W x H. This also applies to the legend.
Line 190: instead of the sentence starting ‘Data were collected...’ Try the following: "The exploration trials and social response assay were recorded and videos analysed using open-source software ‘BORIS’ (Friard and Gamba 2016).”
Line 198: ‘assay’ not ‘essay’
Line 228: remove ‘for later analysis’.
Line 354: I find some confusion with Tables 2, S4. Here you say that you check for the interaction ‘Days in captivity*test type’ and ‘Experience of captivity*test type’. You say you did this in the univariate model and didn’t include them in the multivariate models (because non sig) but both these interactions are in the multivariate results in table 2. Why exactly did you need to do the univariate analysis ?
Line 393 and 394: These numbers in the text do not match table 2. Make sure absolutely everything aligns correctly.
Line 395 and 396: these numbers match table s4 so don’t need to make reference to table 2.
Line 399: Probably worth putting a full stop after captivity in the middle of this line and starting a new sentence beginning "Furthermore, birds with no...”.
Line 402: Again, numbers do not match any of the tables. Double check. If you mean to be referring to table S6, don’t mention table 2.
Line 405: in Table 2, the baseline is the cage whereas everywhere else that test type is mentioned, the baseline is the room. Is there a reason for this? Better to be consistent if there is no specific reason.
Line 406-407: numbers don’t match again.
Line 410: missing reference to table 2.
Line 415: it’s 0.12 rather than 0.13 in the table... Check the numbers.
Table 2: You say the results are based on two multivariate models, so where do the fixed and random effect estimates in this table come from? Model 1 or model 2? I understand that the fixed and random effect estimates for model 2 are shown in table s6.. so where do the fixed effects in table 2 come from?
You say ’intercept coefficients’ after Table S4 on line 6 but do you mean interactions? I’m still unclear why table s4 is needed because everything seems to be shown in table 2.
You've got days in captivity under the model 1 heading here in table 2 but in table s4 you've got it under the fixed effects. Why this difference?
Figure 1: Figure 1 is much improved from the first version so thankyou to the authors for that. Note that in the pdf version, figure 1 and 2 are swapped around. I assume you want the graphs to be figure 2 and the illustrations of the methods as figure 1. When submitting to the journal, be careful of this.
You’ve got the L W and H here on the third line of the legend so make sure you include this format elsewhere in the text and keep the same format.
Line 4 of the legend: add in ’of the room’ after saying ’on the floor’.
Line 6 of legend: you say h for 150cm but nothing for the 40cm. If this is width put w.
Line 11 of legend: from the text, the sliding door is pulled out by 10cm so make that clearer here.
Table S3: you could put the percentages that the PCAs represent. E.g PC1 (41%) etc. And then state in legend what this is.
Line 468: Add in that its the juveniles you mean to be more flexible. ”...rely more on previous experience (refs), so young birds may be more flexible and continue to...”

Check the R code again. Line 323 you say Table S3 when I think you mean Table S4.

---

## Round 0.3 · accepted · Accept

Really interesting study of a phenomenon that is often overlooked, ‘gambit of prior experience’. In this case, prior experience of handling and captivity, in birds.